# Targeting Modifiable Risks: Molecular Mechanisms and Population Burden of Lifestyle Factors on Male Genitourinary Health

**DOI:** 10.3390/ijms26199698

**Published:** 2025-10-05

**Authors:** Xingcheng Yang, Meiping Lan, Jiawen Yang, Yuyi Xia, Linxiang Han, Ling Zhang, Yu Fang

**Affiliations:** 1Department of Clinical Medicine, School of Medical, Wuhan University of Science and Technology, Wuhan 430065, China; yangxingcheng@wust.edu.cn (X.Y.); hanlinxiang@wust.edu.cn (L.H.); 2Department of Environmental Hygiene and Occupational Medicine, School of Public Health, Wuhan University of Science and Technology, Wuhan 430065, China; 19585424574@163.com (M.L.); yangjiawen@wust.edu.cn (J.Y.); xiayuyi@wust.edu.cn (Y.X.)

**Keywords:** health risk factor, suboptimal lifestyle, male health, urogenital system

## Abstract

Health represents a state of complete physical, mental, and social well-being, with lifestyle factors accounting for approximately 60% of health determinants. Suboptimal health describes an intermediate condition between wellness and disease. According to 2023 WHO data, infertility affects approximately 17.5% of global adults, with male factors implicated in 30–50% of cases, establishing infertility as a critical public health challenge. Substantial preclinical and clinical evidence links suboptimal lifestyles to male reproductive dysfunction, positioning these behaviors as modifiable infertility risk factors encompassing environmental contaminants and lifestyle patterns. This systematic review synthesizes evidence on five key lifestyle determinants—tobacco, alcohol, microplastics, sedentariness, and sleep disruption—affecting male genitourinary health. Adopting an evidence-based medicine framework, we integrate epidemiological and experimental research to establish foundational knowledge for developing novel preventive strategies targeting male suboptimal health.

## 1. Introduction

Male reproductive health is a critical component of overall well-being, with profound implications for individual quality of life, family planning, and broader societal health [1]. It encompasses not only physiological function but also psychological and social dimensions, making it a central issue in men’s health. The genitourinary system encompasses organs such as the kidneys, bladder, urethra, prostate, testes, and penis, each playing critical roles in filtration, excretion, reproduction, and hormonal regulation. Key measurable outcomes reflecting their function include renal function markers (e.g., glomerular filtration rate, creatinine clearance), semen parameters (sperm count, motility, morphology), hormonal profiles (testosterone, luteinizing hormone, prostate-specific antigen), and urinary metrics (volume, proteinuria, voiding frequency). However, global male reproductive health indicators are concerning: WHO data indicates approximately 15% of reproductive-aged couples experience infertility, with male factors contributing 20–70% [2]. Over the past 50 years, significant declines in human sperm quality have occurred, with parameters decreasing over 50%, alongside rising clinical incidence of oligozoospermia, asthenozoospermia, and teratozoospermia [3]. Concurrently, the burden of urological diseases is increasing; urological malignancies (e.g., prostate, bladder cancer) rank highly in global male cancer morbidity and mortality, presenting risks across the lifespan [4]. Crucially, paternally inherited abnormalities can impact offspring health via epigenetic mechanisms [5]. Addressing the men’s health crisis demands systemic strategies, from urgent medical intervention to ensuring long-term socio-demographic sustainability, making it a priority research and policy area.

Substantial evidence indicates that lifestyle factors constitute a major, potentially modifiable determinant of men’s health outcomes, often outweighing the contribution of healthcare access itself [6]. Classified by ICD-11 and public health bodies, a ‘suboptimal lifestyle’ denotes a deviation from equilibrium, encompassing nutritional imbalance (e.g., high-fat, low-carbohydrate diets), physical inactivity (>8 h sedentary/day), sleep deficiency (<6 h/day), chronic psychological stress, and addictive behaviors (tobacco/alcohol dependence) [7]. Research confirms that traditional and novel environmental pollutants synergize with modern lifestyles to impair male genitourinary function [8]. Crucially, unlike relatively immutable genetic factors, the modifiable nature of suboptimal lifestyles presents a pivotal intervention target for addressing the men’s health crisis. Consequently, implementing primary prevention through lifestyle medicine is urgently needed to catalyze a paradigm shift from reactive treatment towards proactive health promotion.

## 2. Cigarette Smoking: Dose-Dependent Urogenital Toxicity and Spermatogenic Impairment

Tobacco smoke delivers a complex mixture of toxicants (e.g., nicotine, Cd, Pb) with established gonadotoxic and carcinogenic effects, exhibiting marked male predominance in global prevalence (25% males vs. 5.4% females) [9], particularly in China (26.7–27.2% male smokers) [10]. Clinically, smoking demonstrates a dose/time-dependent association with urological pathology (Figure 1). Current smokers face a 17–22% increased risk of kidney stones and a 9–34% elevated risk of chronic kidney disease (CKD), with prenatal/childhood exposure conferring the highest CKD risk (HR = 1.34, *p* < 0.001) [11]. Male smokers exhibit accelerated renal decline, including a 24% higher CKD progression risk, 5.5–6.9% reduced eGFR and 1.6–2.55-fold increased proteinuria risk (*p* < 0.05) [12,13,14]. For urological malignancies, smoking exhibits a biphasic effect—prostate cancer incidence paradoxically decreases (RR = 0.74, *p* < 0.001) [15] but mortality surges 42% (RR = 1.42, *p* < 0.001) [16], with recent quitters (<15 years) facing a 233% risk escalation (OR = 3.33, *p* < 0.01) [17]. Conversely, bladder cancer risk increases nonlinearly, plateauing at 3.27-fold beyond 20 cigarettes/day and reaching 15.8-fold with ≥20 pack-years (AOR = 15.8, *p* < 0.001) [18,19,20].

Beyond renal pathologies, smoking exerts pronounced dose-dependent damage on male reproductive function, specifically targeting spermatogenesis (Figure 1). Smokers demonstrate significant reductions in sperm concentration, motility, and morphology (reported ORs ranging from 0.84 to 0.89), with heavy smokers (>10 cigarettes/day) showing 13.7% lower concentration and 7.1% reduced progressive motility [21,22,23,24]. Notably, smokeless tobacco users exhibit a 24% decrease in total sperm count despite a 14% testosterone elevation [25], implicating direct gonadal toxicity independent of endocrine disruption. Critically, this reproductive impairment demonstrates reversibility: 3 months of abstinence increases semen volume (16.9%), sperm concentration (22.7%), and total sperm count (44.5%) [26].

These divergent effects reflect tobacco’s dual-phase mechanistic profile. Acute nicotine exposure transiently inhibits tumor growth via neuroendocrine modulation (e.g., dopamine pathways), explaining early paradoxical cancer suppression [27]. Conversely, chronic accumulation of nitrosamines and polycyclic aromatic hydrocarbons drives sustained organ dysfunction, carcinogenesis, and spermatogenic impairment through oxidative stress, DNA damage, and inflammation (Figure 1) [28]. However, antioxidants such as protocatechuic acid (PCA) can mitigate lipid peroxidation and DNA damage by upregulating the expression of antioxidant enzymes including heme oxygenase-1 (HO-1), superoxide dismutase 2 (SOD2), and nitrosylquinoline oxidase 1 (NQO1). Furthermore, by activating the AMPK/PGC-1α/Nrf1 pathway, they improve mitochondrial function, thereby restoring sperm motility and testosterone levels [29]. This biphasic toxicity model unifies the observed clinical spectrum from dose-dependent renal decline and cancer paradoxes to reversible spermatogenic deficits underscoring cessation as a critical intervention point.

Collective evidence indicates that tobacco exposure impairs male reproductive function through multiple pathways, with the severity of impairment exhibiting a dose-dependent relationship. Notably, smoking cessation has been demonstrated to effectively ameliorate sperm parameters.

## 3. Ethanol Exposure: Organ-Specific Pathophysiology and Genetic Polymorphism-Mediated Effects

In 2015, global epidemiological data revealed an average per capita alcohol intake of 6.42 L of pure ethanol equivalents among adult populations [30]. Cross-sectional surveys indicated that 18.3% of adults exhibited binge drinking behavior (operationally defined as single-occasion consumption exceeding 60 g of absolute ethanol) during the preceding 30-day monitoring period [31]. Ethanol is rapidly absorbed in the gastrointestinal tract and distributed systemically, with hepatic metabolism primarily mediated by alcohol dehydrogenase (ADH) and catalase (CAT), oxidizing ethanol to acetaldehyde. This genotoxic intermediate is detoxified by aldehyde dehydrogenase 2 (ALDH2) to acetate, while cytochrome P450 2E1 (CYP2E1) generates reactive oxygen species during microsomal oxidation, collectively establishing pro-inflammatory and oxidative milieus that drive urogenital toxicity prior to renal excretion (Figure 2).

The impact of ethanol exposure on urogenital disorders exhibits significant organotropism and population-stratified susceptibility, mediated through distinct pathophysiological pathways across anatomical compartments (Figure 2). Epidemiologic stratification of renal disease demonstrated a 23% risk reduction for CKD in males with moderate ethanol intake (70 g/d; RR = 0.78) [32]. However, this nephroprotective association underwent complete age-dependent attenuation beyond the sixth decade (RR = 1.00) [32], suggesting metabolic senescence alters ethanol’s pharmacodynamic profile. Paradoxical findings emerged in nephrolithiasis risk assessments. Cross-sectional surveys identified inverse correlations with heavy ethanol consumption (OR = 0.76), whereas Mendelian randomization studies revealed positive associations between drinking frequency (OR = 1.29) and urolithiasis incidence, potentially indicating cumulative metabolic perturbations override ethanol’s transient diuretic effects [33]. Bladder pathophysiology demonstrates critical population stratification patterns. Moderate (RR = 0.97, *p* = 0.59) or heavy (RR = 1.07, *p* = 0.58) alcohol consumption was not associated with the disease in Western populations, yet elevated susceptibility in Japanese populations (RR = 1.31, *p* < 0.01) and spirit-preferring males (RR = 1.42–1.50) [34]. Conversely, drinking ≥10 alcoholic drinks per month was associated with a 59% lower risk of detrusor overactivity (OR = 0.41) [35], which may be mediated by concentration-dependent modulation of bladder smooth muscle γ-aminobutyric acid receptors by ethanol. Prostatic disease analyses demonstrated absence of dose–response relationships between cumulative ethanol–exposure and adenocarcinoma risk (RR = 1.00) [36]. However, developmental exposure assessments identified 3.21-fold elevated odds (OR = 3.21) for high-grade prostate cancer with adolescent ethanol consumption ≥7 sessions/week [37], suggesting androgen-sensitive epithelial plasticity during pubescence. Biochemical surveillance studies documented ethanol-induced suppression of free prostate-specific antigen (fPSA) levels (β = −0.11, *p* < 0.05) and significant ethanol dietary inflammation interaction (*p* = 0.037) [38,39], raising clinical concerns about compromised diagnostic sensitivity in ethanol-consuming populations undergoing PSA-based malignancy screening.

Ethanol exposure manifests multiphasic dose–response relationships with male reproductive pathophysiology, characterized by testicular parenchymal remodeling and spermatozoal quality deterioration (Figure 2). Bilateral testicular atrophy in chronic ethanol consumers, with moderate (Δ = −4.164 mL, *p* < 0.001) and heavy intake cohorts (Δ = −5.500 mL, *p* < 0.001) demonstrating progressive volumetric loss [40]. Frequency-dependent effects were evident with subjects consuming ethanol ≥12 times/week exhibiting 3.451 mL reduced testicular volume versus ≤11 times/month counterparts (*p* < 0.001), suggesting hypothalamic–pituitary–gonadal axis disruption [40]. Meta-analysis of 23,258 hormonal profiles confirmed ethanol-induced endocrine perturbation: suppressed testosterone (SMD = −1.60), follicle-stimulating hormone (FSH; SMD = −0.47), and luteinizing hormone (LH; SMD = −1.35), coupled with elevated estradiol (SMD = 0.22), collectively establishing an anti-spermatogenic steroid milieu [41]. These neuroendocrine alterations directly mediate spermatogenic impairment, evidenced by ethanol-associated declines in seminal volume (Δ = −0.25 mL, 95% CI 0.07–0.42) and morphologically normal spermatozoa (Δ = −1.87%, 95% CI 0.86–2.88). Daily consumers exhibited 5.17% elevated teratozoospermia risk (*p* = 0.03) [42], corroborated by multicenter studies showing concentration-dependent oligozoospermia (OR = 3.72/2.32) [43,44]. Polymorphism-dependent vulnerability was observed in kinetic parameters: ALDH2 allele carriers exhibited precipitous motility decline (43%→20%, *p* = 0.005) post exposure [45], while Han Chinese cohorts demonstrated median total motility reduction (64%→56%, *p* = 0.001) [46]. Nonlinear dose–response patterns emerged in controlled intake scenarios. Dual independent investigations identified J-curve associations with moderate ethanol consumption (4–7 units/week), showing enhanced seminal volume (3.0 vs. 2.4 mL), concentration (31 vs. 24.5 × 10^6^/mL), and total sperm count (87.9 vs. 51.5 × 10^6^/mL) [42,47]. Mechanistic studies implicate oxidative–genotoxic pathways, with murine models exhibiting 52% increased spermatozoal cephalic anomalies (10.5% vs. 6.9%, *p* < 0.01) and accelerated nuclear chromatin decondensation (57.1% vs. 48.3%, *p* < 0.05) [48]. Clinical cohorts demonstrated ethanol dose-dependent sperm DNA fragmentation index elevation (Δ = +5.83%, *p* = 0.002) and corresponding live birth rate reduction (OR = 0.5) [49]. Contradictory findings regarding DNA integrity suggest potential non-oxidative toxicity mechanisms, possibly involving histone modification or spermatozoal miRNA dysregulation [50,51]. Concurrently, alcohol or high-fat diets further induce systemic inflammation and endotoxinaemia by precipitating obesity and gut microbiota dysbiosis. However, the core mechanism ultimately leading to male reproductive dysfunction is oxidative stress damage within the testes, rather than endotoxins themselves [52]. Alcoholic liver injury triggers a systemic inflammatory response through a ‘multiple-hit’ mechanism, allowing pro-inflammatory cytokines (such as TNF-α and IL-6) to infiltrate the testes. This disrupts the blood–testis barrier and suppresses Leydig cell function. Excessive reactive oxygen species (ROS) produced by the liver also induce systemic oxidative stress, collectively causing remote damage to the testes’ hormone synthesis and spermatogenesis environment [53].

Ethanol exerts multifactorial effects on male genitourinary systems, necessitating clinical frameworks that integrate comprehensive assessments of genetic polymorphisms and metabolizing enzyme activities when evaluating toxicological mechanisms and formulating targeted interventions (Figure 2).

## 4. Microplastics and Male Health: Multifaceted Mechanisms, Particle Size-Dependent Effects, and Transgenerational Implications

Microplastics (MPs), defined as plastic particles <5 mm in diameter, are ubiquitously distributed across marine ecosystems, terrestrial soils, atmospheric aerosols, and biological matrices (Figure 3). They predominantly comprise polymers including polyethylene (PE), polystyrene (PS), polyvinyl chloride (PVC), and polypropylene (PP) [54]. Common sources encompass degradation of plastic waste, industrial effluent discharge, and microbeads from personal care formulations. Due to their minute dimensions, high specific surface area, and hydrophobic nature, MPs exhibit a pronounced propensity to adsorb environmental endocrine-disrupting chemicals (EDCs), heavy metals, and pathogenic microorganisms, thereby forming complex composite contaminants [55,56,57,58]. Studies confirm MPs enter organisms via ingestion, inhalation, or dermal absorption, accumulate within organs, and disrupt physiological homeostasis, with reproductive system toxicity being particularly salient.

The mechanistic complexity and key exposure determinants critically govern the reproductive toxicity profile of microplastics. MP-induced reproductive toxicity involves multifaceted pathways, notably oxidative stress, inflammatory cascades, endocrine disruption, and apoptotic signaling. Toxicity exhibits distinct size-dependent variations: microplastics within the size range of 1–5 mm significantly impair sperm motility, viability, and serum testosterone levels by activating the P38 MAPK signaling pathway and inducing oxidative stress, culminating in testicular histopathology [59,60,61]. Microplastics ranging from 1 μm to 5 mm provoke mitochondrial dysfunction, compromise the blood–testis barrier integrity, and induce DNA damage primarily via the NF-κB/Nrf2/HO-1 pathway, leading to diminished sperm count and quality. Microplastics within the size range of 100 nm–1 μm directly inflict structural damage to seminiferous tubules and reduce spermatogenic cell populations [62]. Microplastics smaller than 100 nm, owing to their reduced size, demonstrate enhanced capacity for cellular membrane penetration and testicular accumulation, resulting in more severe reproductive impairment (Figure 3) [63,64,65]. Dose and exposure duration are pivotal determinants; chronic low-dose exposure (e.g., 100 µg/L over 180 days) elicits progressive sperm quality deterioration [66], whereas acute high-dose exposure (e.g., 2000 mg/kg for 28 days) rapidly triggers inflammation and hormonal dysregulation [67]. Dose–response relationships are evident, exemplified by PS-MPs where minimal effects occur at 20 µg/L, escalating to maximal toxicity at 2000 µg/L [68,69,70]. Additives like di(2-ethylhexyl) phthalate (DEHP) and bisphenol A (BPA) potentiate toxicity: DEHP disrupts testicular immune microenvironment homeostasis, promotes Th2/Th17 immune polarization, and inhibits meiotic progression via the Trp53/p38-MAPK pathway [71,72,73,74,75]. BPA impairs germ cell proliferation and reduces sperm quality through activation of the TLR4/NF-κB signaling cascade [76,77,78,79,80]. Furthermore, MP co-exposure with heavy metals (e.g., cadmium) [81], microcystins (MCLR), or arsenic generates synergistic effects [82], amplifying oxidative stress and DNA damage, ultimately manifesting as testicular lesions and increased sperm malformation rates (Figure 3) [83,84,85,86,87].

Experimental models robustly confirm the reproductive toxicity of microplastics. Rodent studies demonstrate MP exposure reduces serum testosterone and gonadotropin (LH, FSH) levels, depletes spermatogenic cells, and compromises blood–testis barrier function [88,89]. Aquatic models (e.g., zebrafish, oysters) exhibit gonadal atrophy, diminished sperm motility, and reduced offspring viability [90,91,92]. Recent animal studies have revealed that 80 nm nanoplastics impair spermatogonial proliferation by inhibiting Cyp26a1, a key gene in retinoic acid metabolism, whilst 5 μm microplastics disrupt energy metabolism by downregulating the thyroid hormone receptor Thra [93]. Clinical manifestations in exposed models include sperm head malformations, reduced motility, testicular fibrosis, and infertility [65,94,95]. While mechanistic insights are emerging, human epidemiological data remain scarce. Future research imperatives include elucidating MP metabolic fate, establishing definitive dose thresholds, and characterizing long-term transgenerational consequences to inform effective protective strategies for vulnerable populations.

## 5. Health Risks Associated with Sleep Disorders in Men: Epidemiological Evidence, Disruption Mechanisms and Clinical Implications

Emerging evidence underscores the critical association between sleep and male reproductive health. However, contemporary epidemiological data reveal a concerning escalation of sleep disorders in male populations. The American Academy of Sleep Medicine categorizes these disorders into six principal classifications: insomnia, sleep-related breathing disorders, central hypersomnia, circadian rhythm sleep–wake disorders, parasomnias, and sleep-related movement disorders (Figure 4) [96]. Longitudinal analyses demonstrate a progressive decline in sleep duration among American males, with mean nightly sleep decreasing from 7.40 to 7.18 h over the past two decades. Notably, the prevalence of short sleepers (≤6 h) has risen significantly from 22.3% to 29.2% during this period [97]. Although temporal stability was observed between 2004 and 2012, the sustained upward trajectory in sleep deprivation rates presents substantial public health challenges.

Recent studies demonstrate distinct organ-specific consequences of sleep disorders within the urinary system. The kidneys are the most frequently affected organs within the urinary system. Individuals maintaining optimal sleep patterns (score = 5) showed 23% lower CKD incidence compared with those with poor sleep hygiene (score ≤ 1) (HR = 0.77, *p* < 0.001) [98]. Additionally, sleep duration demonstrated a nonlinear U-shaped association with CKD risk. Both short (≤6 h) and prolonged sleep (≥8 h) durations increased CKD risk by 13–45% compared to 7 h sleepers (HR = 1.07–1.45; OR = 1.13–1.25), with maximal vulnerability observed at ≤5 h (OR = 1.42) [99]. Mechanistically, sleep deprivation and insomnia directly impaired glomerular filtration capacity through serum creatinine elevation (short sleep: β = 0.14 mg/dL; insomnia: β = 0.09 mg/dL; both *p* < 0.05) [100]. Intriguingly, compensatory sleep strategies demonstrated significant renal protective effects. Weekend sleep extension (>7 h deficit recovery) reduced CKD risk by 56% (OR = 0.44, *p* = 0.03), while systematic napping attenuated sleep debt consequences (β = −0.21 for eGFR decline, *p* = 0.02) [101]. For nephrolithiasis pathogenesis, sleep disturbances followed a cumulative dose–response pattern. Specific impairments—particularly sleep initiation difficulty (OR = 1.68, 95% CI 1.32–2.14), inadequate duration (HR = 1.13 per hour deficit), and poor sleep quality (HR = 1.19 per SD decrease)—independently elevated calculi risk. Each additional sleep disorder compounded the probability by 14–68% through circadian disruption of urinary citrate excretion (*p* < 0.001) [102,103]. In contrast to renal associations, sleep duration showed no significant association with prostate carcinogenesis (RR = 0.88–0.99; OR = 1.18, *p* = 0.427) [104,105]. However, sleep quality degradation significantly predicted benign prostatic hyperplasia progression (β = 0.34 IPSS points per sleep quality unit decline, *p* < 0.001), potentially mediated through autonomic nervous system dysregulation (Figure 4) [106].

Accumulating evidence establishes quantifiable associations between sleep quality and male reproductive parameters through validated Pittsburgh Sleep Quality Index (PSQI) assessments. Multivariable regression analysis reveals that each 1-point elevation in PSQI total score corresponds to significant reductions in semen quality metrics: 9.287-unit decline in total sperm motility (95% CI −12.05 to −6.52), 9.193-unit decrease in forward motility, and 8-unit reduction in concentration [107]. Notably, these sleep-related impairments exhibit compounded effects through dual pathways: insomnia severity demonstrates a dose-dependent inverse correlation with sexual satisfaction (r = −0.167, *p* < 0.01), while concomitant declines in sperm quality parameters synergistically exacerbate reproductive dysfunction [108]. Sleep duration also manifests a U-shaped association with seminal volume (*p* = 0.002) [109]. Both short (<6 h) and prolonged (>9 h) sleep durations reducing ejaculate volume by 12% (95% CI −22% to −0.68%) and 3.9% (95% CI −7.3% to −0.44%), respectively [109]. Mechanistic investigations implicate oxidative stress pathways in this relationship: chronic insomnia correlates with diminished glutathione peroxidase activity (−38% vs. controls, *p* = 0.007) and elevated lipid peroxidation markers (+52%, *p* = 0.003), creating a pro-oxidant microenvironment detrimental to germ cell viability [110]. Pathological sleep disorders exert more pronounced effects. Each unit increase in obstructive sleep apnea (OSA) severity associates with 0.23 SD decrease in sperm vitality (*p* = 0.018) and 0.19 SD reduction in total motility (*p* = 0.025) [103]. Comparative analysis reveals OSA patients exhibit 22.7% lower progressive motility (30.9% ± 23.2% vs. 53.6% ± 11.1%, *p* < 0.001) alongside impaired sexual function, evidenced by 16.7% decline in IIEF scores (25 vs. 30) and 30% reduction in libido (7 vs. 10) [108]. Circadian regulation emerges as a critical modulator. Genetic ablation studies demonstrate Bmal1-deficient mice develop complete infertility (*p* < 0.001), highlighting core clock genes’ reproductive relevance [111]. Clinical data corroborate these findings: subjects maintaining pre-22:30 bedtimes exhibit 2.75-fold higher odds of normal sperm quality (95% CI 1.1–7.1), whereas post-midnight sleepers show significantly diminished sperm counts (−41%, *p* = 0.012) [112]. Environmental chronodisruptors like light pollution may potentiate these effects through circadian desynchronization [113]. Notably, sleep-sperm DNA integrity relationships follow U-curve dynamics: extreme sleep durations associate with 5% elevation in DNA fragmentation index (95% CI −1 to 13%) and 6% increase in free testosterone (95% CI 0–13%) [114]. Although sleep-onset latency prolongation correlates with 33% semen volume reduction (3.0→2.0 mL, *p* = 0.045), multivariate analysis indicates stronger effects on motility parameters (β = 0.78) versus secretory functions (β = 0.32) [115].

The findings of these studies have provided substantial evidence to support the hypothesis that effective sleep management is a critical component of male reproductive health. However, Sleep deprivation disrupts circadian rhythms, activating the hypothalamic–pituitary–adrenal axis. Elevated glucocorticoids inhibit the hypothalamus’s release of GnRH, thereby suppressing pituitary gonadotropin and testicular testosterone synthesis. Concurrently, reduced melatonin secretion due to sleep disturbances diminishes its antioxidant and protective effects, further exacerbating testicular dysfunction [116]. The maintenance of sleep duration at 7–8 h and the enhancement of sleep quality have emerged as pivotal strategies in the prevention of renal diseases. Concurrently, lifestyle modifications, integrative medicine approaches, and the application of emerging technologies have demonstrated the potential to mitigate the adverse effects of sleep disorders on male reproductive health (Figure 4). However, given the heterogeneity in the sensitivity of different organs to sleep disorders, the development of targeted intervention strategies is imperative.

## 6. Profiles of Adverse Outcomes in Men Under Sedentary: Burden of Genitourinary Diseases and Threat to Life Expectancy

Sedentary behavior [117], operationally defined as wakeful activities characterized by low energy expenditure (≤1.5 metabolic equivalents) in seated or reclined postures, has become a pervasive health challenge in contemporary populations with marked demographic disparities. Sedentary metabolic derangements stem from multisystem pathophysiological interactions: diminished skeletal muscle contractile frequency induces functional impairment of lipoprotein lipase (LPL), leading to a 31% reduction in postprandial triglyceride clearance efficacy (*p* < 0.001) [118]. Prolonged sitting contributes to obesity by reducing energy expenditure and promoting visceral fat accumulation. Obesity subsequently triggers hormonal imbalances, such as decreased testosterone levels and increased, estrogen thereby impairing sperm production and male fertility [119]. The primary manifestations of sedentary behavior on male health encompass the urinary system, reproductive function, and mortality.

Prolonged sedentary behavior exerts detrimental effects on urinary system homeostasis, with particular pathophysiological implications for kidney (Figure 5). Canadian cohort data stratified by baseline kidney function demonstrate a dose-dependent relationship between sitting time and renal impairment risk [120]. Individuals with advanced renal insufficiency (eGFR < 45 mL/min/1.73 m^2^) exhibit 4.2-fold elevated risk (95% CI 2.5–7.3), while moderate renal dysfunction (eGFR 45–60 mL/min/1.73 m^2^) shows 1.7-fold risk elevation (95% CI 1.2–2.3). This gradient pattern persists across ethnic groups, as evidenced by accelerated renal decline in US Hispanic/Latino populations (−0.06% eGFR reduction per sedentary hour; 95% CI −0.10 to −0.02) [121]. Notably, sex-specific susceptibility patterns emerge from multinational datasets. Female subjects exceeding 8 h daily sitting thresholds demonstrate significantly greater chronic kidney disease progression than males (*p* < 0.01) [122]. Conversely, Korean epidemiological surveys reveal tentative associations between physical inactivity and renal dysfunction in adult males, though lacking statistical significance (*p* > 0.05) [123]. These divergent profiles suggest potential gender dimorphism in sedentary-induced nephropathy mechanisms. When evaluating broader urological outcomes, nonlinear associations become apparent. Comparative analysis reveals a U-shaped relationship between sitting duration and nephrolithiasis risk: 34.1% risk reduction occurs at 6–8 h/day (OR = 0.659, 95% CI 0.457–0.950) versus < 6 h, yet transitions to 12.3% elevation beyond 8 h (OR = 1.123), albeit nonsignificant (*p* = 0.571) [124]. Prostatic inflammation exhibits linear progression, with weekly sitting exceeding 30 h conferring 24% elevated risk (HR = 1.24, 95% CI 1.05–1.45) relative to <1 h controls [125]. Methodological variations in sedentary behavior research—including divergent operational definitions (3–12 h/day thresholds) and measurement tools (self-reports vs. accelerometry)—may explain outcome inconsistencies. Current limitations extend to underpowered cohorts (<10,000 participants in 83% studies) and incomplete confounder adjustment (activity patterns, sociocultural factors), compromising causal attribution. To bridge these gaps, we advocate: standardized metrics incorporating metabolic biomarkers; multinational cohorts (N ≥ 10,000) with sex-stratified designs; and risk-adapted exercise protocols for vulnerable subgroups (e.g., women with incipient renal decline). These priorities aim to decode biological mechanisms and advance precision prevention frameworks.

The association between sedentary behavior and male fertility also remains controversial in current research (Figure 5). While population-level evidence indicates no statistically significant direct correlation between sedentary lifestyle and infertility risk (adjusted OR = 1.20, 95% CI 0.55–2.61, *p* = 0.63), emerging data suggest potential indirect pathways mediated through metabolic alterations. Notably, individuals with elevated adiposity demonstrate a 2.83-fold increased infertility risk (95% CI 1.31–6.10, *p* < 0.01), implying adipose-related metabolic dysregulation may constitute a critical mediating mechanism [126]. Intriguingly, behavior-specific analyses reveal differential impacts: prolonged television viewing (>5 hr/day) associates with 28.8% reduced sperm concentration (37 vs. 52 million/mL) and 34.2% lower total sperm count (104 vs. 158 million), potentially attributable to localized thermal stress or oxidative damage mechanisms [127]. This contrasts with broader epidemiological findings showing no significant correlations between total sedentary duration and conventional semen parameters. Prospective cohort analyses confirm nonsignificant associations for sperm concentration (*p* = 0.37), vitality (*p* = 0.92), and morphology (*p* = 0.16) [128], with subsequent studies replicating these null findings for motility parameters (progressive motility *p* = 0.69; total motility *p* = 0.53) [129]. Notably, a critical interaction emerges between physical activity and sedentarism. Active individuals exhibit superior sperm motility profiles (progressive motility *p* = 0.03; total motility *p* = 0.03) compared to sedentary counterparts, despite comparable sperm concentrations (*p* = 0.37) and total counts (*p* = 0.82) [130]. This dissociation suggests sedentary behavior may preferentially impair functional rather than quantitative aspects of sperm biology, potentially through inactivity-induced metabolic perturbations.

Existing studies demonstrate substantial variations in sedentary-related mortality outcomes across chronic disease subtypes (Figure 5). Our stratified analysis demonstrates differential patterns in CKD populations: while physical activity levels showed no significant inverse correlation with all-cause mortality (*p* > 0.3) [131], prolonged sitting (≥8 h/day) independently elevated mortality risk by 67% (HR = 1.67, 95% CI 1.32–2.11) [132]. Notably, renal function biomarkers (eGFR and urine albumin-to-creatinine ratio [UACR]) emerged as significant predictors in this dose–response relationship. Subgroup analyses revealed distinct protective effects of exercise across clinical populations. Among renal cell carcinoma survivors, high-intensity exercise (≥7 h/week) conferred a 40% mortality reduction (HR = 0.60, 95% CI 0.47–0.76), contrasting with the null association observed for prolonged television viewing [133]. Similarly, patients with hyperuricemia achieving recommended exercise thresholds experienced an 11% lower mortality risk (HR = 0.89, 95% CI 0.82–0.97) [134]. Intriguingly, while sedentary time showed no direct mortality association in prostate cancer patients (*p* = 0.24), sufficient exercisers exhibited an 11% survival advantage over inactive counterparts (ΔHR = 0.89, 95% CI 0.83–0.95), suggesting partial mitigation of sedentarism’s detrimental effects through compensatory physical activity [135].

## 7. Conclusions

Male genitourinary health faces escalating threats from a complex matrix of non-traditional factors, including environmental contaminants and behavioral risks. These agents heighten disease susceptibility through distinct yet interconnected pathways: epigenetic toxicity, DNA fragmentation, hormonal suppression, hyperthermia, and circadian rhythm dysregulation. Interactions between factors range from antagonism to synergistic potentiation, complicating risk prediction. Individual vulnerability varies significantly due to genetic predispositions and heterogeneous environmental exposures across the life course. Critically, certain exposures induce transgenerational harm via epigenetic mechanisms, compromising not only the exposed individual but also offspring phenotypes. This underscores the imperative to address modifiable risks to mitigate reversible damage and intergenerational health crises.

Despite deepening scientific understanding, the actual spectrum of subclinical exposures encountered in real-world settings is vastly broader and more intricate. Addressing this complexity necessitates focused research advances in several critical directions: First, elucidating the mechanisms underlying the combined effects of multi-factorial exposures and their interactions with behavioral co-factors. Second, developing and validating targeted intervention strategies that leverage windows of reversibility to mitigate or repair damage. Third, establishing integrated high precision risk assessment frameworks that synthesize exposomic data, large-scale cohort analyses, and validated individual biomarkers. Fourth, prospectively investigating the long-term and transgenerational health consequences of parental exposures on offspring. Only through such multidimensional and systems-oriented research can evidence-based preventive paradigms and effective clinical interventions be robustly developed to safeguard male genitourinary health.

## Figures and Tables

**Figure 1 ijms-26-09698-f001:**
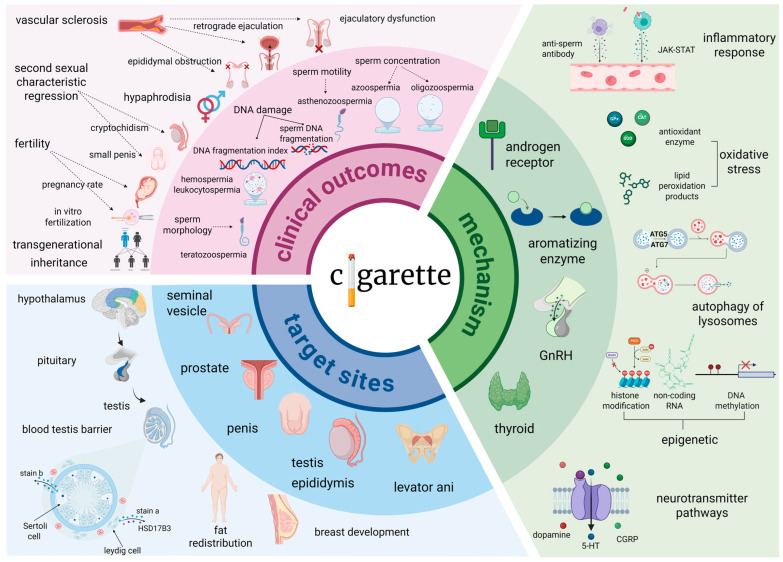
Male smoking and fertility risk: pathophysiological mechanisms and clinical evidence. Clinical outcomes manifesting as reduced sperm quality, sexual dysfunction, impaired fertility, and intergenerational risks. Affected sites encompass the testes, epididymis, prostate, and endocrine system. Key mechanistic pathways involve hormonal disruption, inflammation, oxidative stress, dysregulated autophagy, and epigenetic modifications. GnRH, Gonadotropin-Releasing Hormone; CGRP, Calcitonin Gene-Related Peptide; JAK-STAT, Janus Kinase-Signal Transducer and Activator of Transcription; ATG5, Autophagy-related gene 5; ATG7, Autophagy-related gene 7; HSD17B3, Hydroxysteroid 17-beta dehydrogenase 3; 5-HT, 5-Hydroxytryptamine; SAM, S-Adenosyl methionine; PRC2, Polycomb Repressive Complex 2; RNAPⅡ, RNA Polymerase II; Me3, Trimethylation.

**Figure 2 ijms-26-09698-f002:**
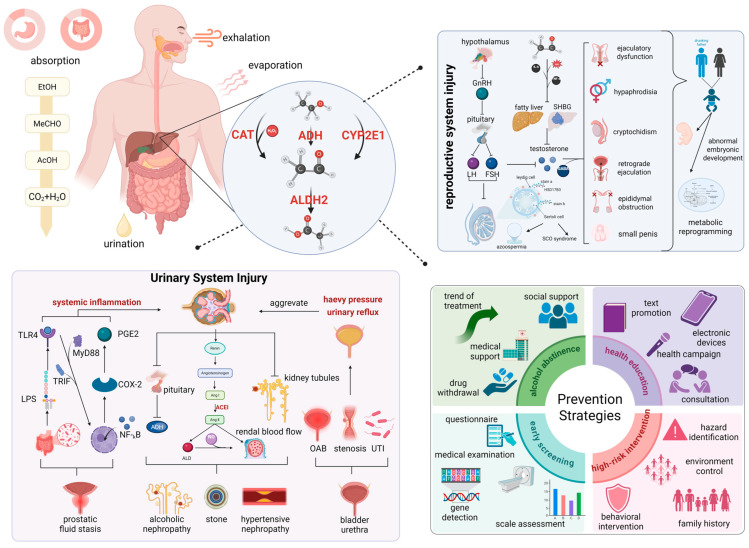
Alcohol exposure: multi-organ pathological damage in men. Urological sequelae encompass renal ischemic damage, electrolyte imbalance, bladder detrusor dysregulation causing overactivity, heightened urethral infection/stricture risk, and prostatic reactions. Reproductive impacts include sexual dysfunction, azoospermia, diminished testosterone, and offspring developmental disorders with metabolic reprogramming via epigenetic mechanisms. Preventive strategies include controlled intake, hydration, biomarker surveillance, and early high-risk group intervention. CAT, Catalase; ADH: Antidiuretic Hormone; SHBG, Sex Hormone-Binding Globulin; GABA, Gamma-Aminobutyric Acid; SCO, Sertoli Cell-Only Syndrome; OAB, Overactive Bladder; UTI, Urinary Tract Infection; ACEI, Angiotensin-Converting Enzyme Inhibitor; LPS, Lipopolysaccharide; CYP2E1, Cytochrome P450 2E1; ALDH2, Aldehyde dehydrogenase 2; ALD, Aldehyde dehydrogenase; ADH, Alcohol dehydrogenase; TLR4, Toll-like receptor 4; TRIF, TIR-domain-containing adapter-inducing interferon-β; MyD88, Myeloid differentiation primary response 88; PGE2, Prostaglandin E2; COX-2, Cyclooxygenase-2; NF-kB, Nuclear factor kappa-light-chain-enhancer of activated B cells.

**Figure 3 ijms-26-09698-f003:**
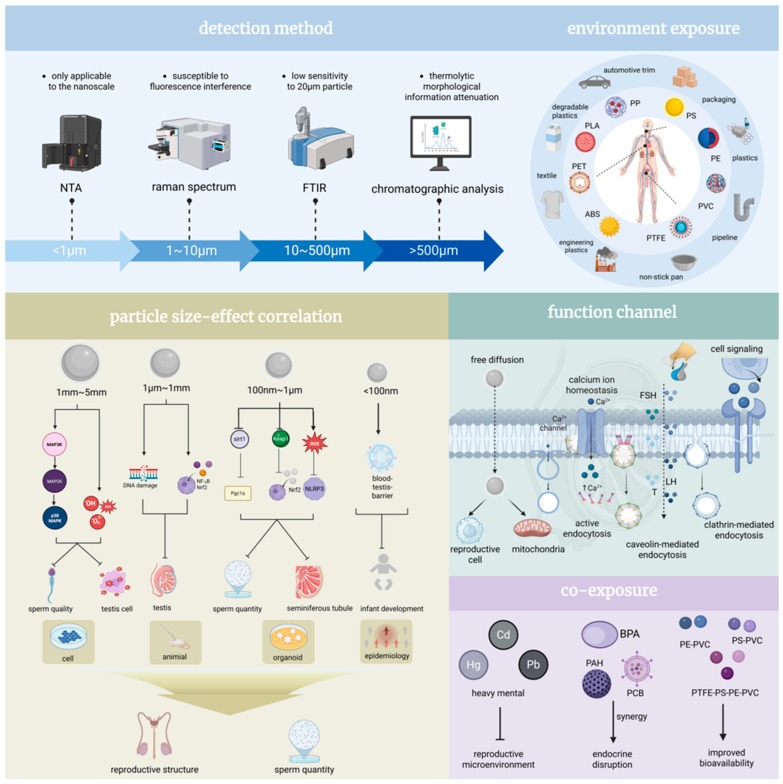
Size-dependent impacts of microplastics on male urogenital health. Human exposure pathways associated with diverse microplastic types; detection methodologies applicable across varying particle size fractions, coupled with intrinsic strengths and limitations; size-dependent associations between microplastic properties and male health outcomes, alongside the mechanisms; and synergistic interactions resulting from concurrent exposure to microplastics and co-contaminants. NTA, Nanoparticle Tracking Analysis; FTIR, Fourier Transform infrared spectroscopy; Pgc1α, Peroxisome proliferator-activated receptor γ coactivator 1-α; Nrf2, Nuclear Factor erythroid 2-Related Factor 2; NLRP3, NOD-like receptor thermal protein domain associated protein 3; PP, Polypropylene; PS, Polystyrene; PE, Polyethylene; PVC, Polyvinyl chloride; PTFE, Polytetrafluoroethylene; ABS, Acrylonitrile butadiene styrene; PET, Polyethylene terephthalate; PLA, Polylactic acid; PP, Protein phosphatase; MAP2K, Mitogen-activated protein kinase kinase; MAP3K, Mitogen-activated protein kinase kinase kinase; P38, p38 Mitogen-activated protein kinase; MARK, MAP/microtubule affinity-regulating kinase; NF-fB, Nuclear factor kappa-light-chain-enhancer of activated B cells; sir1, Silent information regulator 1; keap1, Kelch-like ECH-associated protein 1; ROS, Reactive oxygen species; T, Testosterone; BPA, Bisphenol A; PAH, Polycyclic aromatic hydrocarbo; PCB, Polychlorinated biphenyl; PE-PVC, Polyethylene-Polyvinyl chloride; PS-PVC, Polystyrene-Polyvinyl chloride; PTFE-PE-PVC, Polytetrafluoroethylene-Polyethylene-Polyvinyl chloride.

**Figure 4 ijms-26-09698-f004:**
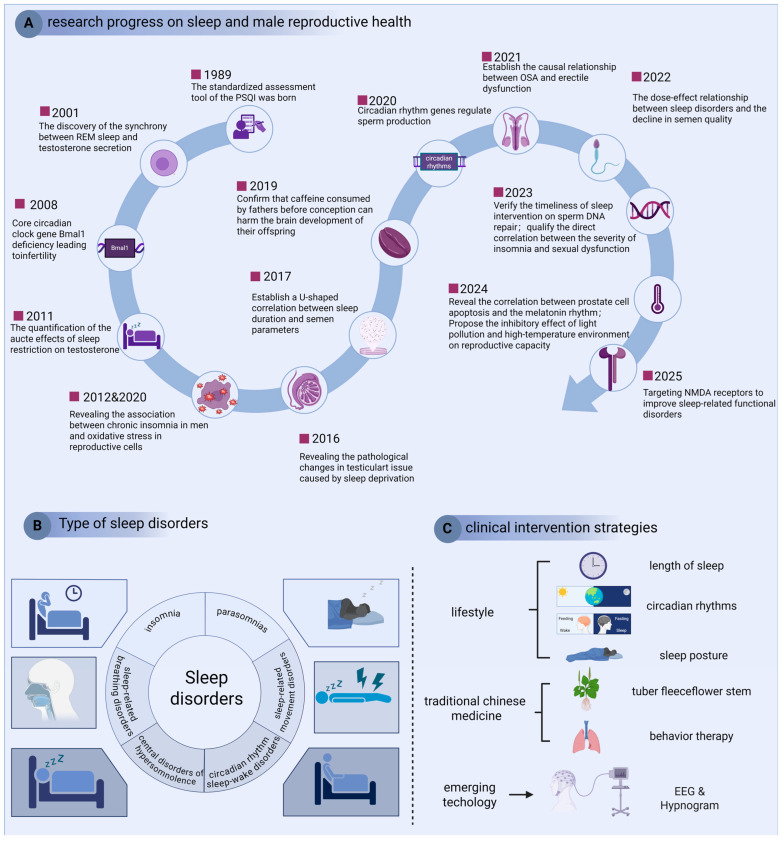
Response of male health to sleep disorders. Research over the past 25 years have indicated melatonin secretion, light rhythm and intensity, adenosine metabolism, and body temperature regulation mediating sleep regulation of male health; the classification of sleep disorders defined by the American Academy of Sleep Medicine (AASM) ranges from insomnia disorder to sleep-related movement disorders; and clinical interventions are mainly based on lifestyle. SQI, Pittsburgh Sleep Quality Index; REM, Rapid eye movement; OSA, Obstructive sleep apnea; NMDA, N-Methyl-D-aspartic acid; EEG, Electroencephalogram.

**Figure 5 ijms-26-09698-f005:**
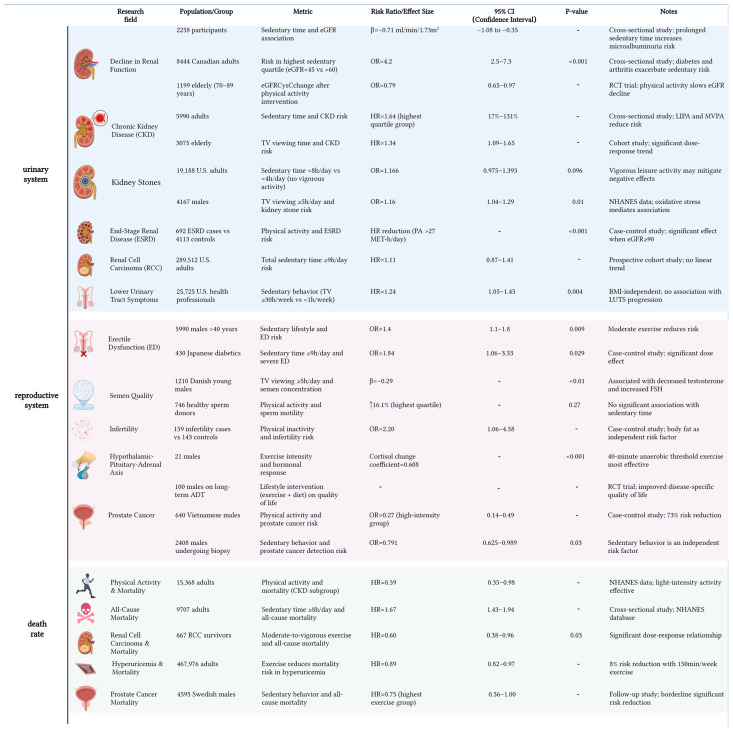
Epidemiological burden of sedentary behavior on male health. Summarized from urinary system, reproductive system and mortality. Effects of sedentary behavior across different races and age-stratified males were investigated based on pooled cohort analyses. The observed adverse outcomes encompass pathologies spanning the entire genitourinary tract and demonstrate elevated mortality risk. These findings were robustly demonstrated to be statistically significant across multiple parameters. CKD, Chronic kidney disease; ESRD, End-stage renal disease; RCC, Renal cell carcinoma; ED, Erectile dysfunction; ADT, Androgen deprivation therapy; RCC, Renal cell carcinoma; eGFR, Estimated glomerular filtration rate; RCT, Randomized controlled trial; LIPA, Lysosomal acid lipase; MVPA, Moderate to vigorous physical activity; NHANES, National Health and Nutrition Examination Survey; BMI, Body mass index; LUTS, Lower urinary tract symptoms; FSH, Follicle-stimulating hormone.

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
