# Peer review of "Targeting Modifiable Risks: Molecular Mechanisms and Population Burden of Lifestyle Factors on Male Genitourinary Health"

_ijms, 2025, doi:10.3390/ijms26199698_

Round 1

Reviewer 1 Report

Comments and Suggestions for Authors
  1. The first three sentences of the introduction are somewhat redundant. It would be more effective to directly state the importance of male fertility issues.
  2. The manuscript overlooks an important factor, namely obesity. Male obesity, caused by dietary factors such as high-fat or high-sugar diets, has a significant impact on semen quality. The authors should incorporate this perspective, referencing studies such as (DOI: 10.1002/fft2.484).
  3. A significant issue throughout the manuscript is that, despite each point being supported by substantial evidence, including epidemiological data, the discussion is limited to stating these points. The mechanisms by which these lifestyle factors lead to reduced male fertility and the pathways through which they exert their effects are not addressed.
  4. The subheadings chosen by the authors are somewhat misleading. While such subheadings might be appealing as titles for a paper, they hinder readability and comprehension within the manuscript.
  5. The authors should note that male genitourinary health is a broad topic, yet the review primarily focuses on male fertility. It is recommended that the authors revise the introduction, title, and abstract to focus the core discussion on male fertility.
  6. In the section on microplastics, the use of maternal cases is inappropriate. As noted in point 4, subheadings do not need to be overly novel. The authors should clearly elucidate the mechanisms by which microplastics affect semen quality without citing inappropriate cases, such as those related to transgenerational effects, for the sake of novelty.
  7. Overall, this is a highly valuable and significant topic, likely to garner high citation rates upon publication. The manuscript cites numerous epidemiological and cohort studies, but all proposed lifestyle factors lack discussion of direct mechanisms. It is recommended that the authors supplement this, with potential approaches including: the toxicity of substances in tobacco to the testes and sperm; the impact of alcohol-induced alcoholic fatty liver, multiple hits, systemic inflammatory factors, and oxidative stress on the testes; the reproductive toxicity of microplastics; and the effects of sleep disorders on endocrine dysregulation and the hypothalamic-pituitary-testicular hormone balance. A separate paragraph discussing mechanisms could be added for each point. Additional references can be expanded to include not only human studies but also in vitro studies and research on model animals highly relevant to human reproduction (e.g., DOI: 10.1186/s40104-024-01031-6).

Reviewer 2 Report

Comments and Suggestions for Authors

In this review, the Authors examine the impact of tobacco use, alcohol consumption, microplastic exposure, sedentariness, and sleep disruption on male genitourinary health, synthesizing epidemiological and experimental evidence within an evidence-based medicine framework to highlight their role as modifiable risk factors for infertility and suboptimal health. The topic is timely and relevant, and the manuscript addresses an important gap by linking lifestyle factors with male reproductive and urological outcomes.

However, I recommend major revisions before the manuscript can be considered further.

Specific comments:

  1. The Authors should better justify the rationale for focusing on these specific determinants. It remains unclear whether the choice was guided by prevalence, strength/quality of evidence, novelty, or public health relevance.
  2. It would be important to introduce a dedicated paragraph on the determinants of genitourinary health. Specifically, the review should outline which organs are involved and which measurable outcomes (e.g., semen parameters, renal function markers, hormonal profiles) best reflect their function. This would provide readers with a conceptual framework to better contextualize the subsequent analysis of lifestyle-related risks.
  3. The figures are not sufficiently integrated into the text. The narrative should clearly explain and contextualize what each figure illustrates. In their current form, they appear confusing and do not fully support the discussion.

Round 2

Reviewer 1 Report

Comments and Suggestions for Authors

The author has made careful revisions to the paper and it can now be accepted in its current form.

Author Response

Thank you for your recognition.

Reviewer 2 Report

Comments and Suggestions for Authors

The Authors have addressed all my concerns and I have no further comments. 

As far as I am concerned, the manuscript is now acceptable to be published.

Author Response

Thank you for your recognition.